# Isotretinoin promotes elimination of translation-competent HIV latent reservoirs in CD4T cells

J. Natalie Howard[1], Callie Levinger[1], Selase Deletsu[1], Rémi Fromentin[2], Nicolas Chomont[2], Alberto Bosque[1]*, for the AIDS Clinical Trials Group (ACTG) A5325 Team

1 Department of Microbiology, Immunology, and Tropical Medicine, George Washington University, Washington DC, United States of America, 2 Centre de recherche du CHUM et Département de microbiologie, infectiologie et immunologie, Université de Montréal, Montréal, Canada

* abosque@gwu.edu

**Data Availability Statement:** All relevant data are within the manuscript and its Supporting information files.

## Abstract

Development of novel therapeutic strategies that reactivate latent HIV and sensitize reactivated cells to apoptosis is crucial towards elimination of the latent viral reservoir. Among the clinically relevant latency reversing agents (LRA) under investigation, the γc-cytokine IL-15 and the superagonist N-803 have been shown to reactivate latent HIV *ex vivo* and *in vivo*. However, their clinical benefit can be hindered by IL-15 promoting survival of infected cells. We previously identified a small molecule, HODHBt, that sensitizes latently infected cells to death upon reactivation with γc-cytokines through a STAT-dependent pathway. In here, we aimed to identify and evaluate FDA-approved compounds that could also sensitize HIV-infected cells to apoptosis. Using the Connectivity Map (CMap), we identified the retinol derivative 13-*cis*-retinoic acid (Isotretinoin) causes similar transcriptional changes as HODHBt. Isotretinoin enhances IL-15-mediated latency reversal without inducing proliferation of memory CD4 T cells. *Ex vivo* analysis of PBMCs from ACTG A5325, where Isotretinoin was administered to ART-suppressed people with HIV, showed that Isotretinoin treatment enhances IL-15-mediated latency reversal. Furthermore, we showed that a combination of IL-15 with Isotretinoin promotes the reduction of translation-competent reservoirs *ex vivo*. Mechanistically, combination of IL-15 and Isotretinoin increases caspase-3 activation specifically in HIV-infected cells but not uninfected cells. Our results suggest that Isotretinoin can be a novel approach to target and eliminate translation-competent HIV reservoirs.

## Author summary

Even with the development and rapid advancement of antiretroviral therapies (ART) against HIV in the past three decades, there is still no generalized cure for HIV. Identification of novel therapeutic interventions that target the latent reservoir could provide new avenues to find a cure. In this work, we evaluated the ability of the FDA-approved

**Funding:** This work was supported by the National Institutes of Health grants (R21/R33AI116212 and R56AI145683 to AB), and the NIH Institute of Allergy and Infectious Disease (UM1AI068636 to the ACTG A5325 cohort). This research has been facilitated by the services and resources provided by District of Columbia, Center for AIDS research, an NIH funded program (AI117970), which is supported by the following NIH co-funding and participating institutes and centers: NIAID, NCI, NICJD, NHLBI, NIDA, NINH, NIA, FIC, NIGGIS and NDDK and OAR. The content is solely the responsibility of the authors and does not necessarily represent the official views of the NIH. The funders had no role in study design, data collection and analysis, decision to publish, or preparation of the manuscript.

**Competing interests:** The authors have declared that no competing interests exist.

compound Isotretinoin to reactivate latent HIV and promote a reduction of the latent reservoir *ex vivo*. Our studies found that Isotretinoin can enhance the ability of IL-15 to reactivate latent reservoirs. Furthermore, using samples from clinical trial ACTG A5325 (NCT01969058) we show that Isotretinoin sensitizes latent HIV to reactivation by IL-15. Finally, we demonstrate that Isotretinoin can sensitize reactivated cells to death via apoptosis. Overall, this study demonstrate that Isotretinoin could be used in combinatorial strategies towards the development of an HIV cure to promote reduction of latent reservoirs.

## Introduction

Even with the development and rapid advancement of antiretroviral therapies (ART) against HIV in the past three decades, there is still no generalized cure for HIV. This is due to the presence of latent viral reservoirs that are harbored not only within different subsets of CD4+ T cells [1–7], but also macrophages [8, 9], $\gamma\delta$ T-cells [10], microglia, astrocytes, and dendritic cells [11–15]. These latent reservoirs remain unseen by the immune system, which allows for viral persistence despite therapy [4, 16, 17]. In spite of this seemingly insurmountable hurdle, to date there are several cases of HIV cure, primarily within patients who received bone marrow or stem cell transplants from CCR5Δ32 donors for coinciding blood cancers [18–21], and most recently from a donor heterozygous for the CCR5Δ32 mutation [22]. These cases are the hallmark of HIV eradication, where all replication-competent virus is eliminated from the body [23]. However, cell transplants are not a universally applicable cure strategy for the 38 million people globally living with HIV. To that end, the development of a therapeutic intervention with the potential to be implemented worldwide is needed.

A major cure strategy under investigation is the "shock-and-kill" or "kick-and-kill" approach, which depends on transcriptional activation of latent HIV proviruses with the subsequent killing of reactivated cells either by immune effector cells or the inherent cytopathic effects of HIV. This strategy uses a small molecule generally termed latency reversing agent (LRA) and has the potential to be scalable to the larger population. However, to date several LRAs have undergone clinical evaluation and none of them have shown significant measurable reductions in the size of the reservoir [24, 25], so it is imperative to identify new effective strategies. One promising therapeutic avenue for the "shock" component with potential widespread clinical utility is the use of the common gamma chain ($\gamma$c)-cytokine interleukin-15 (IL-15) and the cognate drug N-803. IL-15 and N-803 function via the Janus-associated kinases (JAK)/ signal transducer and activator of transcription (STAT) pathway and results in phosphorylation of STAT5. IL-15 signaling and subsequent STAT5 activation have been shown to promote the survival and proliferation of HIV-specific cytotoxic T cells and natural killer (NK) cells [26, 27], and to stimulate HIV latently infected cells into actively producing viral transcripts *in vitro* [28, 29]. N-803 is an IL-15 receptor superagonist complex that has shown strong and broad anti-cancer effects in several murine cancer models [30, 31], as well as enhanced tissue retention compared to soluble IL-15 [32]. Multiple clinical trials evaluating the effectiveness of N-803 for treatment of various cancers have been carried out, as well as a phase I trial in people with HIV (PWH) for safety and potential LRA efficacy [31, 33–36]. It was observed in the trial that administration of N-803 was well tolerated across a range of doses. In terms of its function as an LRA, a transient increase in viral transcripts per million memory CD4s was observed over the time of administration, but by 6 months after the final dose, viral transcript levels were on average back at the pre-dose baseline. Additionally, there

was no clear impact of N-803 on the size of the latent reservoir, with only a mild increase in intact proviral DNA copies after completion of administration, and no changes in the populations of cells harboring 3' or 5' defective provirus [36]. These results suggest that the future use of N-803 as an LRA is promising, but hinges on finding compounds that are able to enhance its activity *in vivo*.

Recently, an increased resistance to apoptosis of latently infected cells has been proposed as a mechanism of HIV persistence [37–44]. In these prior studies, antiapoptotic proteins of the B-Cell CLL/Lymphoma 2 (Bcl-2) family such as BCL2 and BCL2 Like 1 (BCL2L1 or BCLXL) [37, 38, 40] or inhibitor of apoptosis (IAP) family member Baculoviral IAP Repeat Containing 5 (BIRC5) [39] have been shown to protect latently infected cells from apoptosis. A study evaluating simian immunodeficiency virus (SIV) persistence identified elevated levels of the caspase-8 inhibitor CASP8 And FADD Like Apoptosis Regulator (CFLAR or CFLIP) in cells harboring viral DNA *in vivo* [45]. As such, there has been growing interest in advancing cure strategies that induce apoptosis pathways within reactivated cells to deplete the latent reservoir. Recent work has found that inhibition of the pro-apoptotic protein BCL-2 and the RNA helicase DDX3 trigger selective apoptosis in HIV-infected cells over uninfected bystanders [37, 40, 46–48].

We have previously described the small molecule 3-Hydroxy-1,2,3-Benzotriazin-4(3H)-one (HODHBt) as a LRA that enhances $\gamma$c-cytokine signaling by sustaining phosphorylation and transcriptional activity of STAT5 through inhibition of the nonreceptor type phosphatases PTPN1 and PTPN2 [49, 50]. HODHBt treatment increases IL-2 and IL-15-mediated activation of STAT5, promoting reactivation from latency in a primary cell model and cells isolated from PWH [51]. Additionally, HODHBt reduces the size of the inducible latent reservoir in a primary cell model of latency [49], suggesting that it can sensitize infected cells to death. Furthermore, we have recently shown that HODHBt can also enhance IL-15 mediated NK and CD8T cell effector function [52, 53]. To accelerate the clinical translation of these results, we used the Connectivity Map (CMap) to identify FDA-approved compounds that could promote similar transcriptional changes and could behave similar to HODHBt [54]. Using this tool, we identified and confirmed that the FDA-approved compound Isotretinoin can enhance the LRA activity of IL-15 and specifically promote the elimination of HIV-infected cells through the induction of a pro-apoptotic program.

## Materials and methods

### Materials

Human recombinant interleukin-2 (rIL-2) and recombinant interleukin-15 (rIL-15) were obtained via the BRB/NCI Preclinical Repository. Human $\alpha$IL-12 (500-P154G), $\alpha$IL-4 (500-P24) and TGF-$\beta$ (100–21) were purchased from Peprotech. $\alpha$CD3/CD28 DYNAL Dynabeads (#11132D) purchased from Invitrogen. FDA compounds were purchased from Selleckchem (Isotretinoin, #S1379; Trimethobenzamide, #S5483, Clotrimazole, #S1606, Bumetanide, #S1287, Lomustine, #S1840). Raltegravir (#HRP-11680) and Nelfinavir (#ARP-4621) from NIH HIV Reagent Program. CellTrace Yellow proliferation kit (#C34567) was purchased from Invitrogen. CytoTox 96 non-radioactive cytotoxicity assay (#G1780) was purchased from Promega. Cell Meter Live Cell Caspase 3/7 Binding Assay Kit (#20101) was purchased from AAT Bioquest. Counting beads were from Life Technologies (C36950). Antibodies were purchased from BioLegend (APC-Cy7 $\alpha$CD69 #310914, FITC $\alpha$CD4 clone RPA-T4 #300538, BV605 $\alpha$CD56 clone HCD56 #318334, AF700 $\alpha$CCR7 clone G043H7 #353243, APC-Cy7 $\alpha$CD27 clone O323 #302815), BD Biosciences (PerCP-Cy5.5 $\alpha$CD3 Clone SP34-2 #552852, BV786 $\alpha$CD3 Clone SP34-2 #563800), eBiosciences (eF450 fixable viability dye, FITC $\alpha$CD95 clone

DX2 #11-0959-42, APC $\alpha$CD4 clone OKT4 #17-0048-42, PE $\alpha$CD8 clone OKT-8 #12-0086-42, PerCP-Cy5.5 $\alpha$CD45RA clone HI100 #45-0458-42), Beckman Coulter (FITC KC57 #6604665), Cell Signaling Technologies (STAT5- #94205S, pSTAT5 (Y694)- #9359S, PTP1B- #5311S, TCPTP (TC45)- #58935S, GAPDH- #5174S, HSP90- #4877S, Mcl1- #4572S, BCL2- #15071S, BCLxL- #2764S, Noxa- #14766S, Bim- #2933T), Sigma Aldrich ($\beta$-actin (AC-15)- #A5441), and Jackson ImmunoResearch ($\alpha$Rabbit 2° #111-035-046, $\alpha$Mouse 2° #115-0350146).

## Methods

**Ethics statement.** Participants and samples: Central memory CD4+ T cells were generated from blood samples collected from 17 years and older HIV-negative donor volunteers (Gulf Coast Regional Blood Center). White blood cell concentrates (buffy coat) prepared from a single unit of whole blood by centrifugation were purchased. For the experiments aimed at measuring the effects of isotretinoin on the translation-competent reservoir *ex vivo*, cells collected from 12 PWH on ART were used. All participants were adults and signed informed consent forms approved by the McGill University Health Centre and the Centre Hospitalier de l'Université de Montréal review boards. All participants were male living with HIV under suppressive ART for at least 3 years. They underwent leukapheresis to collect large numbers of PBMCs. PBMCs were isolated by Ficoll density gradient centrifugation and were cryopreserved in liquid nitrogen. CD4+ T cells were enriched by negative magnetic selection using the EasySep Human CD4 T Cell Enrichment Kit (StemCell Technology, 19052).

Study Approval and Design: A5325 was a double-blind randomized controlled multicenter trial of the effects of Isotretinoin therapy on peripheral and gut T cell immune activation, systemic inflammation and change in viral reservoir among chronically HIV-1 infected participants on ART (clinicaltrials.gov identifier NCT01969058). Participants were PWH on suppressive therapy with HIV-1 RNA <75 copies/ml for the preceding 12-month period. Because of the potential teratogenic risk of isotretinoin, all female participants who were of reproductive potential were excluded. Deidentified PBMCs samples at pre-entry and 16 weeks in both the no study treatment arm and the Isotretinoin treatment were obtained to evaluate the presence of translation-competent reservoirs.

Each ACTG A5325 clinical research site had the A5325 protocol and consent forms, approved by their local Institutional Review Boards (IRB) (University of Alabama at Birmingham IRB; University of North Carolina Office of Human Research Ethics; University of Cincinnati IRB; Cone Health IRB; Partners Human Research Committee (now Mass General Brigham IRB); Lifespan IRB; UCLA IRB; University of Rochester Research Subjects Review Board; and Vanderbilt University IRB) as well as registered with and approved by the NIH Division of AIDS Regulatory Support Center Protocol Registration Office (Bethesda, Maryland, USA), prior to any participant recruitment and enrollment. Once a participant for study entry was identified, details were carefully discussed with the prospective participant by clinical staff at the site. The participant (or, when necessary, the parent of legal guardian in the participant was under guardianship) was asked to read and sign the ACTG-approved protocol consent form. PBMCs from aviremic participants from ACTG A5325 were obtained through a collaboration with ACTG and used *in vitro* only. Aviremic participants are anonymous to us at George Washington University, and we did not recruit new participants for these studies. All the samples used had been collected and we have a Non-Human Subject Research determination from the George Washington University.

**Connectivity Map (CMap).** Gene IDs of differentially expressed genes from CD4 and NK cell treated with HODHBt RNAseq data sets (CD4: 222 upregulated genes, 172 downregulated genes; NK: 155 upregulated genes, 47 downregulated genes) were transformed into Affymetrix

probe names (AFFY HG U133A probes). Probe files were saved as .grp files, one for upregulated genes and one for downregulated genes. These .grp files were uploaded into the Connectivity Map 02 tool with outputs of both detailed and permuted results (S1 Data). Instances are defined as compounds that have more than one hit across cell lines and/or treatment concentrations in the reference database.

**Generation of latently infected $T_{CM}$.** Naïve CD4+ T cells were isolated via negative selection from PBMCs obtained from HIV negative donors. Cultured $T_{CM}$ were generated and infected as previously described [55–57]. Naïve CD4 T cells were isolated from PBMCs from HIV-negative donors by negative selection (Stemcell #19555) and activated at $0.5x10^6$ cells/mL with $\alpha$CD3/CD28 Dynabeads (1:1 bead-to-cell ratio) in the presence of 1$\mu$g/mL $\alpha$IL-4, 2$\mu$g/mL $\alpha$IL-12, and 10ng/mL TGF-$\beta$ for 72 hours. Dynabeads were removed on day 3 and cells were subsequently expanded in RPMI supplemented with 1% L-Glutamine, 10% Fetal Bovine Serum, and 1% penicillin/streptomycin (complete RPMI) with 30 IU/ml IL-2 before being infected on day 7 via spinoculation with the X4-tropic virus NL4-3. Levels of intracellular p24 were assessed 72 hours later (day 10) by flow cytometry prior to the infected cells being crowded in 96-well round bottom plates to facilitate spread of infection (100,000 cells/well). On day 13, the cells were uncrowded and plated in the presence of an ART cocktail (1$\mu$M Raltegravir + 0.5$\mu$M Nelfinavir) and 30IU/ml IL-2, and p24 levels were again measured by flow cytometry. 96 hours later (day 17), the CD4 positive cells were sorted from the infected cultures by positive selection (Dynabeads CD4 positive Isolation kit, Thermo Fisher Scientific #11331D), and p24 levels were measured pre- and post- sort. The CD4 positive cells were then resuspended at $1x10^6$ cells/mL and plated with reactivation conditions for a further 48 hours and reactivation was measured by p24 stain on day 19.

For the depletion assay, samples were treated as previously described [49] with the modifications of 100ng/mL IL-15 in place of IL-2, and the addition of 10$\mu$M Isotretinoin.

**Proliferation assay.** Memory CD4 T cells were isolated from HIV-negative donor PBMCs using the CD4 T Cell Enrichment Kit (StemCell #19052). Cells were then resuspended at $1x10^6$ cells/mL in complete RPMI and treatment conditions plus CellTrace Yellow Proliferation dye following manufacturer's instructions for 7 days. Cells were stained for viability and analyzed by flow cytometry to measure cell proliferation.

**Clinical trial samples.** PBMCs ($\sim 25x10^6$ cells per participant per time point) were thawed in 20mL media + 4$\mu$L Benzoase nuclease (1x). Cells were counted and resuspended at a density of $3.3x10^6$ cells/mL in the presence of ART (1$\mu$M Raltegravir + 0.5$\mu$M Nelfinavir). $0.5x10^6$ cells were taken for phenotyping analysis by flow cytometry. The remaining cells were treated with the reactivation conditions of interest and plated in 6-well plates for 96 hours. 96 hours later, supernatants were collected for p24 analysis and $1x10^6$ cells were collected for activation analysis by flow cytometry. Levels of p24 in the supernatants were measured as previously described [58]. Supernatant p24 concentrations for each experimental well were obtained from Quanterix software as [pg/mL] and were converted to [fg/mL] for subsequent analyses. Wells with non-quantifiable p24 values were assigned values of 3 fg/mL and denoted as open circles. Averages for all wells of one reactivation condition were calculated and plotted. The average of all the experimental limits of detection (LODs) across assays was calculated and graphed (aLOD) + 2 standard deviations to ensure that any value above this threshold has a 95% chance of being a true positive. Raw p24 values were normalized to $10^6$ plated PBMCs for each participant/ timepoint/ stimulation condition.

**TRAnslation-CompEtent viral Reservoir (TRACER) assay.** PBMCs ($100x10^6$ cells per donor) were thawed in 10mL media + 2$\mu$L Benzonase nuclease (1x). Cells were counted and total CD4 T cells were isolated via negative selection (StemCell #19052). After isolation, cells were counted and resuspended in complete media at a concentration of 750,000 cells/ 150$\mu$L

media in the presence of ART ($1\mu$M Raltegravir + $0.5\mu$M Nelfinavir). Cells were treated with the reactivation conditions of interest and plated in 96-well round bottom plates ($150\mu$L/ well, 2–3 wells/ reactivation condition) for 96 hours. 96 hours later, $\alpha$CD3/$\alpha$CD28 wells were resuspended and the plate was placed on a plate magnet for 1 min. Cell suspensions were moved to an empty row to remove beads. Plates were then spun down at 2000rpm for 5 mins, $4°$C and supernatants were transferred to a fresh plate. Cell pellets were lysed using $75\mu$L NETN supplemented with protease and phosphatase inhibitors per well in the round bottom plate. Both supernatant and lysate plates were sealed with plate covers and stored at $-80°$C. Prior to p24 analysis, 1% triton was added to supernatants to ensure viral inactivation. Lysate plates were thawed on ice for 30min and spun down at 4000rpm for 30min at $4°$C to remove non-soluble material. Cleared lysates were transferred to a new plate. Levels of p24 in cell supernatants and extracts were measured as previously described [58].

Supernatant p24 concentrations for each experimental well were obtained from Quanterix software as [pg/mL] and were converted to [fg/mL] for subsequent analyses. Wells with non-quantifiable measurements were assigned values of 1 fg/mL. Averages for all wells of one reactivation condition were calculated and plotted. The average of all the experimental LODs across assays was calculated and graphed (aLOD). The average HIV negative baseline (aHNB) was calculated by taking the average of all wells for all the reactivation conditions + 2 standard deviations to ensure that any value above this threshold has a 95% chance of being a true positive.

Lysate p24 concentrations for each experimental well were obtained from Quanterix software as [pg/mL] and converted to [fg/mL]. With the remaining $25\mu$L of lysates for each well, a 1:10 dilution in ddH2O was done followed by a standard BCA to obtain protein concentration in ($\mu$g/$\mu$L). Protein concentration was multiplied by the volume/well used in the p24 assay to obtain an average amount of protein loaded in each ELISA well. The p24 values in fg/mL were then divided by the total protein loaded to obtain a normalized measure of p24 (fg/mL)/ $\mu$g protein. The average values for all wells of one reactivation condition were calculated and graphed. For wells that had non-quantifiable p24 measurements, but where protein was measured via BCA, a value of 0.1 p24 (fg/mL)/$\mu$g was assigned. The aHNB for the lysates was calculated as described above.

To evaluate toxicity of reactivation conditions in the TRACER assay, $12.5\mu$L of each well was transferred to a fresh 384-well flat-bottom plate. Next, $12.5\mu$L of prepared CytoTox 96 reagent was added to each well, and the plates were incubated at room temperature for 30 min in the dark. Finally, $12.5\mu$L of stop solution was added to each well, and the absorbance was recorded using a spectrophotometer at 490 nm.

**Flow cytometry.** To assess surface expression of CD69 and CD95, $5 \times 10^5$ were stained with $0.1\mu$L of viability dye (eBioscience fixable viability dye eFluor 450) in $100\mu$L of phosphate-buffered saline (PBS) for 10 min at 4C. The cells were then stained with $0.5\mu$L of $\alpha$CD69- APC-Cy7 and $0.5\mu$L of $\alpha$CD95- FITC in $100\mu$L of PBS plus 3% FBS for 30 min at $4°$C. For clinical trial sample phenotyping, $0.5\times10^6$ cells were taken per participant per time point and stained with viability dye, $\alpha$CD3-BV786, $\alpha$CD4-FITC, $\alpha$CD8-PE, $\alpha$CD45RA-PerCP/ Cy5.5, $\alpha$CCR7-AF700, and $\alpha$CD27-APC-Cy7. For clinical trial sample activation, $0.5\times10^6$ cells were taken per participant per time point per stimulation condition and stained with viability dye, $\alpha$CD3-PerCP/Cy5.5, $\alpha$CD4-FITC, $\alpha$CD8-PE, $\alpha$CD56-BV605, and $\alpha$CD69-APC-Cy7. To analyze reactivated cells and caspase activation, cells were stained for CD4, viability, and intracellular p24-Gag as previously described [56]. All experiments were run on a BD LSR Fortessa X20 flow cytometer with FACSDiva software (Becton Dickinson, Mountain View CA). Data was analyzed using FlowJo (TreeStar, Inc., Ashland, OR).

**Western blotting.** To measure levels of pro- and anti-apoptotic proteins after treatment with Isotretinoin, uninfected day 10 primary CD4s from latency model cells were treated with the indicated conditions for 24 hours. Cells were then washed with PBS and lysed with NETN extract buffer comprised of 100mM NaCl, 20mM Tris-Cl (pH 8), 0.5mM EDTA, 0.5% Nonidet P-40, protease inhibitor cocktail (cOmplete, Roche), and phosphatase inhibitor cocktail (phosSTOP, Roche) for 30 minutes on ice. Lysates were purified by centrifugation at 12,000 rpm for 10 minutes at 4°C and proteins were visualized on SDS-PAGE. All primary antibodies used at 1:1000 concentrations except for $\beta$-actin (1:10,000). Secondary anti-rabbit and anti-mouse antibodies were used at a 1:10,000 dilution.

**Caspase activation assay.** NL4-3-infected CD4 T cells on day 10 from latency model were resuspended at $1\times10^6$ cells/mL in complete RPMI + 30IU/mL IL-2 and crowded in 96-well plates for 48 hours at 100,000 cells/well. On day 12, the cells were uncrowded and resuspended at $1\times10^6$ cells/mL plus ART combination ($1\mu$M Raltegravir + $0.5\mu$M Nelfinavir). TF3-DEVD-FMK caspase-3/7 reagent from AAT Bioquest was then added to all cells (1:150) before being split into experimental conditions and treated. Cells were then cultured for an additional 48 hours (day 14) before being stained for intracellular p24 and analyzed via flow cytometry.

**Statistics.** Statistical analyses were performed using GraphPad Prism 9.0 software. The statistical analysis used is indicated in each figure legend. A p value of less than 0.05 was considered significant (\*, p<0.05, \*\*, p<0.01).

## Results

### CMap analysis identifies Isotretinoin as an FDA-approved compound that enhances IL-15-mediated latency reversal in a $T_{CM}$ model of latency

We have previously described that the small molecule HODHBt reactivates latent HIV through inhibition of the nonreceptor type phosphatases PTPN1 and PTPN2 [50]. This inhibition, when combined with concomitant $\gamma$c-cytokine-stimulated STAT5 activation, results in the sustained phosphorylation of STAT5, which binds to the HIV LTR and drives viral transcription [49]. We have also shown that HODHBt reduces the size of the inducible latent reservoir in a primary cell model of latency [49]. However, HODHBt has a high $EC_{50}$ in the micromolar range, reducing its potential clinical utility [50, 51]. For this reason, we were interested in identifying FDA-approved compounds that cause similar transcriptional changes as HODHBt in the hopes of repurposing them as agents to enhance latency reversal and sensitize infected cells to death after reactivation. To identify these compounds, we used the Connectivity Map (CMap) tool [54]. We used previously published RNA sequencing data from primary CD4 T cells and NK cells treated with HODHBt and $\gamma$c-cytokines and compared with 1,309 compounds in 5 unique cell lines [49, 52, 54]. A total of 327 compounds were identified as having similar transcriptional changes to that of HODHBt with multiple instances, and subsequent filtering for compounds based on p-value (p<0.05) resulted in 15 compounds. Further filtering for clinical relevance resulted in five FDA-approved candidates (Fig 1A and S1 Data).

We have previously shown that HODHBt in combination with the $\gamma$c-cytokine IL-2 induces expression of CD69 on CD4 T cells [49] and this expression correlates positively with both levels of pSTAT5 [51] and percentage reactivation in a primary cell model of latency (S1 Fig). As such we tested the ability of the FDA-approved compounds to upregulate CD69 expression on the surface of primary CD4 T cells to first validate the identified compounds. Of the five compounds, only Isotretinoin was able to significantly upregulate CD69 expression in the presence of IL-2 in CD4 T cells (Fig 1B). We next performed a dose response with Isotretinoin, and

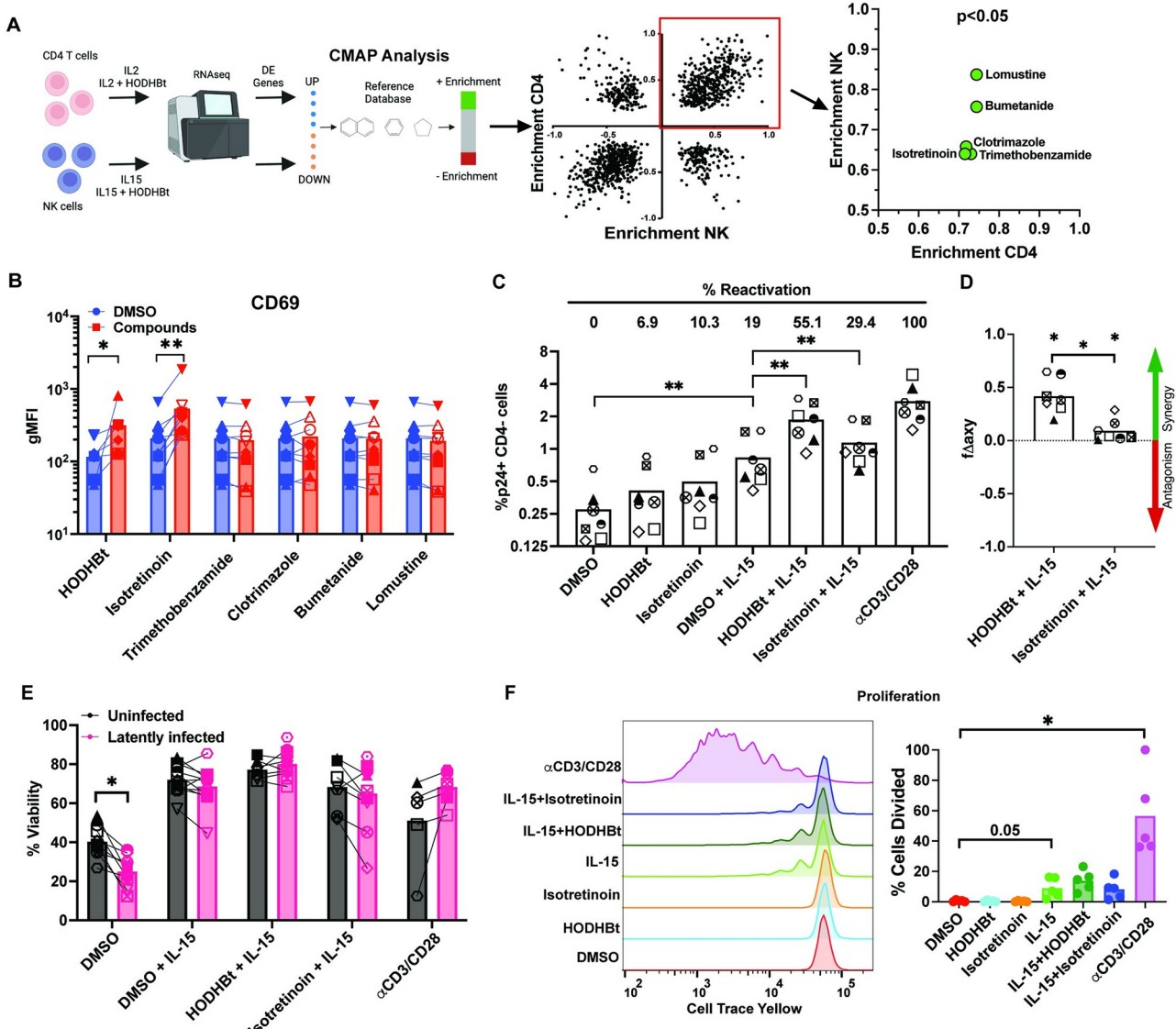

**Fig 1. Identification and confirmation of Isotretinoin as a latency reversal agent.** A: Workflow to identify 5 FDA-approved compounds with transcriptional profiles similar to lead LRA using CMap analysis. Figure created with BioRender. B: CD69 MFI of matched cultured $T_{CM}$ (n = 9) were measured by flow cytometry after treatment with 30 IU/mL IL-2 or IL-2 plus 100$\mu$M HODHBt or 10$\mu$M of indicated compounds for 72 hours. Significant p values over cells treated with IL-2 alone. C: Reactivation of latent HIV in $T_{CM}$ (n = 7) measured by flow cytometry after treatment with 100$\mu$M HODHBt or 10$\mu$M Isotretinoin alone, plus 100 ng/mL IL-15, or $\alpha$CD3/CD28. D: Calculation of synergy for LRA combinations using the Bliss independence model (n = 7). Data is presented as the difference between the observed and expected fractional response. E: Viability of uninfected or latently infected matched conditions (n = 7). F: Memory CD4 T cells were stained with CellTrace yellow and treated with 100$\mu$M HODHBt or 10$\mu$M Isotretinoin alone or in combination with 100 ng/mL IL-15, or $\alpha$CD3/CD28 for 7 days. Proliferation was measured by flow cytometry (n = 5). Wilcoxon matched-pairs signed rank test was used to calculate p values (*p < 0.05; **p < 0.01).

while the $EC_{50}$ was defined in the mid nanomolar range (59nM) (S2A Fig), we observed the highest activity in the absence of toxicity at 10$\mu$M (S2B Fig). For this reason, we subsequently moved forward with this concentration for all the following experiments.

The potential of Isotretinoin as a LRA was first evaluated in the cultured $T_{CM}$ model of HIV latency [55, 57]. Both HODHBt and Isotretinoin had limited LRA activity on their own, consistent with our previous studies demonstrating that HODHBt needs to be combined with a

γc-cytokine for activity (Fig 1C). Reactivation with IL-15 alone resulted in a significant increase compared to the DMSO control, 19% of the positive control αCD3/CD28. This is consistent with previous work indicating that IL-15 is a mild LRA [28, 51]. The addition of HODHBt to IL-15-treated samples resulted in significantly increased reactivation to 55.1% compared to IL-15 alone while reactivation with IL-15 with Isotretinoin resulted in an increase to 29% of the maximal stimulus (Fig 1C). Using the Bliss independence model, reactivation of both compounds with IL-15 was calculated to be significantly synergistic, and combination of IL-15 with HODHBt was significantly more synergistic than combination of IL-15 with Isotretinoin (Fig 1D). Reactivation was observed without any significant changes in viability between the infected and uninfected matched cultures (Fig 1E). Because of the established ability of IL-15 to enhance T cell proliferation, we addressed whether HODHBt or Isotretinoin further enhances IL-15-dependent expansion of memory CD4 T cells. Memory CD4 T cells were isolated from uninfected donor peripheral blood mononuclear cells (PBMCs), stained with CellTrace yellow to measure proliferation, and treated with HODHBt and Isotretinoin alone or in combination with IL-15 for 7 days. We observed an increase in the percent of dividing cells between DMSO and IL-15 conditions, which was expected given the known proliferative effects of IL-15 [59]. However, we observed no increase in proliferation of the compounds alone, or in combination with IL-15 (Fig 1F). This data shows that Isotretinoin enhances IL-15-mediated HIV reactivation in primary cells without exacerbating proliferation of primary memory CD4 T cells.

## Isotretinoin administration enhances IL-15 reactivation susceptibility *ex vivo*

The recently completed clinical trial ACTG A5325 (NCT01969058) was designed to assess the *in vivo* effects of Isotretinoin treatment on enhancing CD4 counts in ART-suppressed PWH with incomplete CD4 reconstitution. To evaluate the impact of Isotretinoin administration for 16 weeks on the latent reservoir, we performed a phenotypic characterization of PBMCs from ten placebo and Isotretinoin-treated participants at entry (pre) and after 16 weeks of treatment (post). No significant changes in either the CD4/CD8 ratio or the naïve and terminally differentiated CD4 T cell subsets between pre- and post-treatment time points were observed for either the placebo or Isotretinoin groups (S3, S5A and S5B Figs). In the placebo group, a significant increase in the central memory CD4 T cell subset was measured between the pre- and post-treatment timepoints. This corresponded to a significant decrease in the transitional memory and effector memory CD4 subsets. A non-significant increase in the central memory CD4 population was observed in the Isotretinoin group (p = 0.06), but no significant changes were observed in the other CD4 cell subsets (S3 and S5C Figs).

To further investigate the impact of Isotretinoin administration, we evaluated the responsiveness of cells isolated from these participants to IL-15-mediated immune activation and latency reversal. PBMCs were stimulated with either IL-15 or αCD3/28 and levels of activation were assessed by flow cytometry after 96h of stimulation (S4 Fig). Both IL-15 and αCD3/28 promoted immune activation of CD4+ T cells, CD8+ T cells and CD56+ NK cells to similar extent in both the placebo and Isotretinoin groups both at pre and post treatment (Fig 2). In the placebo arm, we observed a significant increase in the frequency of CD4+ T cells expressing CD69 between the pre- and post-treatment timepoints in the unstimulated condition (p = 0.012) and a significant decrease in frequency of CD8+ T cells expressing CD69 after IL-15 stimulation (p = 0.03), but these changes were not seen in the Isotretinoin arm (Fig 2A). In general, no significant changes were observed in the cells from people receiving Isotretinoin

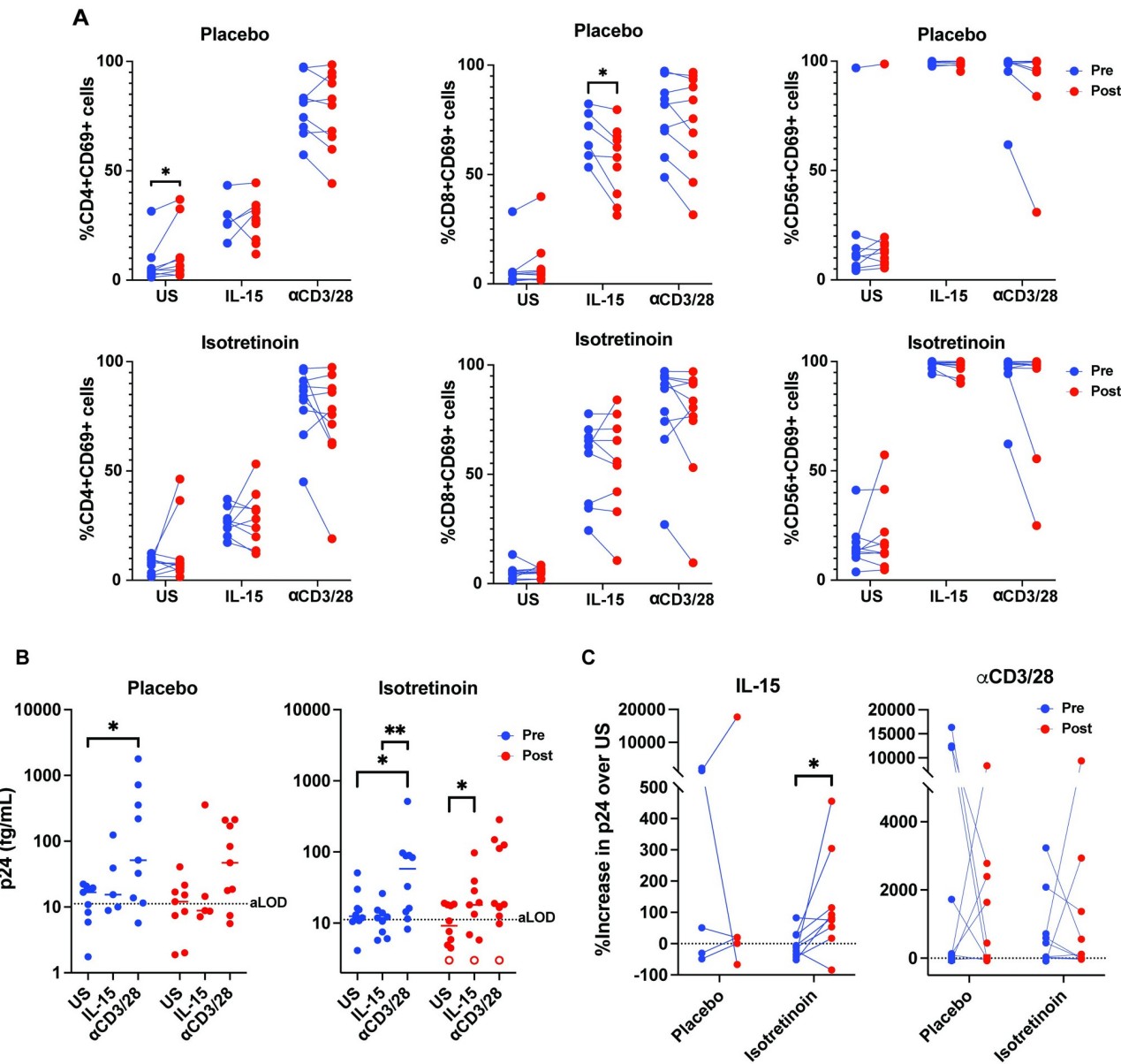

**Fig 2. Effects of Isotretinoin on latent viral reactivation.** A: CD4, CD8, and NK cell activation analysis after stimulating with indicated conditions for 96 hours (n = 10 per arm). B: p24 levels were measured in the supernatants of PBMCs (blue = pre-Isotretinoin treatment, red = post-16 weeks of isotretinoin treatment) left unstimulated or treated with either 100 ng/mL IL-15 or αCD3/CD28 for 96 hours. Open circles indicate participants where the raw p24 values were non-quantifiable as described in the methods. Dashed line denotes the assay average limit of detection (aLOD) and median values are shown. C: % Increase in released p24 over the unstimulated condition for both IL-15 and αCD3/CD28 reactivation conditions. Wilcoxon matched-pairs signed rank test was used to calculate p values (*p < 0.05; **p < 0.01).

compared with placebo suggesting that Isotretinoin treatment *in vivo* does not interfere with IL-15 or αCD3/28-induced immune activation *ex vivo*.

We next evaluated whether Isotretinoin administration *in vivo* influences CD4+ T cells to latency reversal by IL-15 or αCD3/28 *ex vivo*. 96 hours post-stimulation, cell supernatants were collected for p24 Gag measurement using an ultrasensitive planar array p24 Gag ELISA [58]. In the placebo arm we observed a significant increase in reactivation in the pre-treatment

group between the unstimulated and $\alpha$CD3/28-stimulated conditions (p = 0.04). This significant increase was also observed in the Isotretinoin arm pre-treatment group (p = 0.02), as well as between the IL-15 and $\alpha$CD3/28-stimulated conditions (p = 0.008). A significant reduction in relative CD4 counts per $10^6$ PBMCs between the pre- and post-treatment timepoints was seen for the placebo arm (p = 0.008) but was not observed in the Isotretinoin arm (S5E Fig). Interestingly, exposure to Isotretinoin *in vivo* lead to a significant increase in the ability of IL-15 but not $\alpha$CD3/28 to reactivate translation-competent reservoirs *ex vivo* (Fig 2B and 2C).

## Isotretinoin reduces the translation-competent latent viral reservoir in cells from PWH

Given our above findings, we sought to assess whether latent viral reactivation by concomitant Isotretinoin and IL-15 stimulation similarly leads to enhanced reactivation in CD4T cells isolated from ART-suppressed PWH. Total CD4 T cells from 12 ART-suppressed participants and 2 HIV negative participants were isolated and treated for 96 hours with HODHBt or Isotretinoin in the absence or presence of IL-15. Gag p24 protein was measured in both the supernatants and cell lysates and cytotoxicity of the conditions was measured in the supernatants using a lactate dehydrogenase (LDH) readout using the TRAnslation-CompEtent viral Reservoir (TRACER) Assay (see Materials and methods). Results were stratified according to whether there was an increase in p24 viral release measured in the supernatants of the $\alpha$CD3/CD28-stimulated condition compared to the unstimulated control. All the participants have detectable total HIV DNA (S6 Fig). For the eight participants that responded to $\alpha$CD3/CD28, minimal reactivation with HODHBt and Isotretinoin was detected. A significant increase in p24 release between the unstimulated conditions and the conditions treated with IL-15 (p = 0.0078) and $\alpha$CD3/CD28 (p = 0.0078) was observed, while the IL-15 with HODHBt and IL-15 with Isotretinoin reactivation conditions were trending significant (p = 0.055 and p = 0.078 respectively). In most participants, we observed a reduction in p24 release between the IL-15 and IL-15 with HODHBt (6/8, p = 0.742) or Isotretinoin (7/8, p = 0.008) conditions (Figs 3A and S7). In the cell lysates of responders, there was significant increase in reactivation compared to unstimulated after treatment with IL-15 (p = 0.016) and $\alpha$CD3/CD28 (p = 0.0078), while treatment with IL-15 and HODHBt and IL-15 with Isotretinoin were trending significant (p = 0.109 and p = 0.055, respectively). As in the supernatants, there was a reduction in p24 production when HODHBt (7/8, p = 0.055) or Isotretinoin (6/8, p = 0.078) were added to IL-15 (Figs 3B and S8). To determine if the reduction in p24 expression in the combination conditions was due to increased toxicity caused by the treatments, we measured the cytotoxicity for each condition. We did not observe any toxicity associated with any of the treatments except for $\alpha$CD3/CD28 reactivation (p = 0.016) (Fig 3C). For the $\alpha$CD3/CD28 non-responders, we did not observe p24 release into the supernatants (Figs 3D, S7 and S8), or in the cell lysates (Figs 3E and S8) for any condition. We also observed no distinct pattern in the cytotoxicity of the reactivation conditions in these donors (Fig 3F). These results suggest that combination of IL-15 with Isotretinoin may lead to a reduction of translation-competent reservoirs by sensitizing reactivated cells to cell death.

## Isotretinoin enhances IL-15-mediated NOXA and FAS expression and sensitizes HIV-infected cells to apoptosis, leading to depletion of the latent viral reservoir

We have previously shown that HODHBt enhances $\gamma$c-cytokine-mediated viral transcription and translation, which leads to depletion of the inducible latent reservoir in primary cells [49]. Given our observations that combined Isotretinoin and IL-15 treatment reduces the

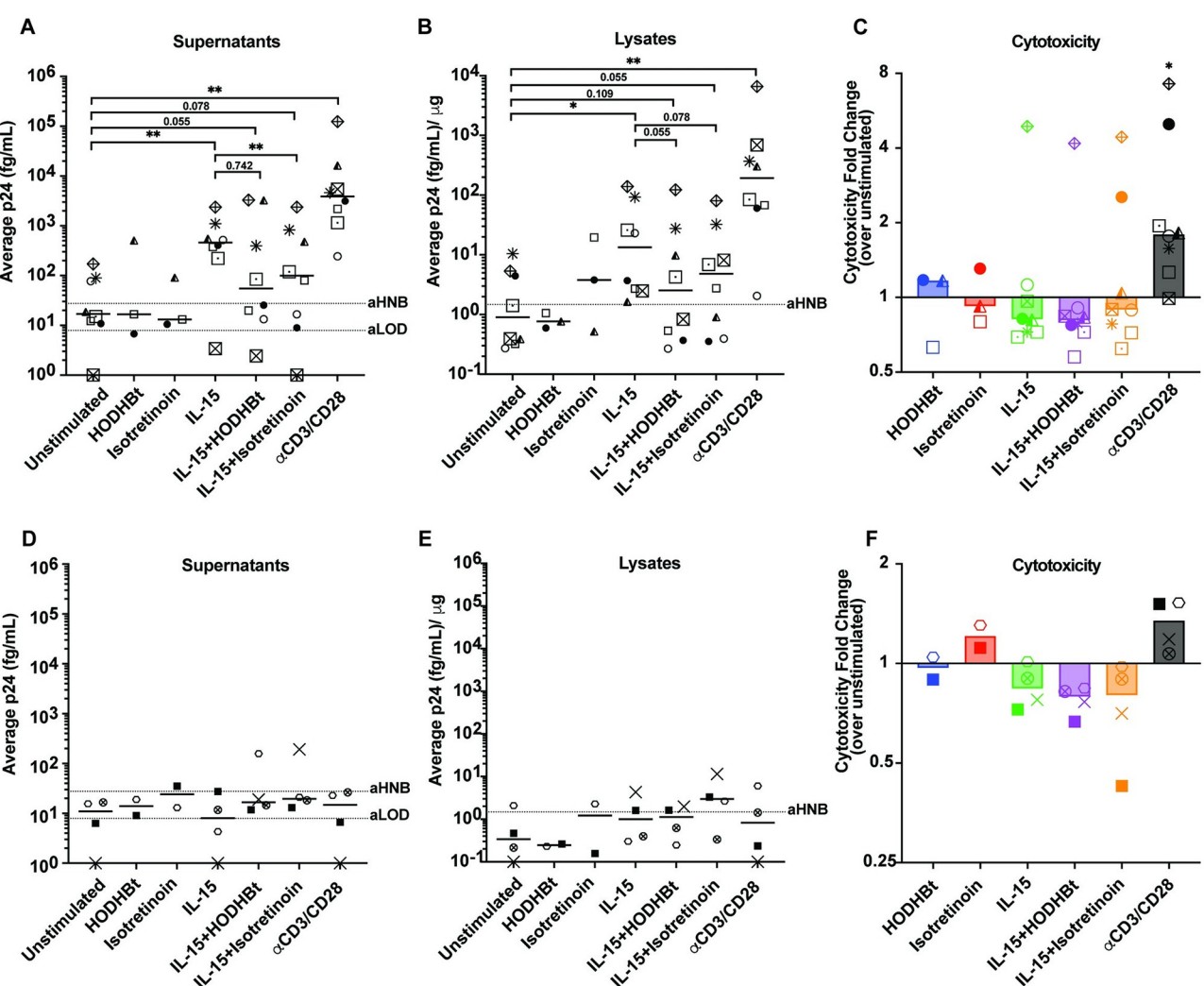

**Fig 3. Effects of Isotretinoin on the translation-competent viral reservoir *ex vivo* in cells from PWH.** P24 levels were measured in supernatants (A) and cell lysates (B) after culture for 96 hours with the labeled conditions (n = 8) for $\alpha$CD3/CD28 responders. For the supernatants and cell lysates, dashed line denotes the HIV-negative baseline (aHNB) and average limit of detection (aLOD). Horizontal lines display median values for each treatment condition. C: Cytotoxicity of treatment conditions for each participant was measured compared to unstimulated controls. P24 levels were measured in supernatants (D) and cell lysates (E) after culture with the labeled conditions (n = 4) for $\alpha$CD3/CD28 non-responders. F: Cytotoxicity of treatment conditions for each participant was measured compared to unstimulated controls. Wilcoxon matched-pairs signed rank test was used to calculate p values (*p < 0.05; **p < 0.01).

translation-competent reservoir in cells from ART-suppressed PWH, we next asked whether this corresponded to a similar depletion in the inducible reservoir. To answer this question, we used a variation of the $T_{CM}$ model of latency as previously described [49]. We observed a significant reduction in the remaining translation-competent viral reservoir after initial reactivation with both HODHBt and IL-15, and Isotretinoin and IL-15 (Fig 4A). This data suggests that Isotretinoin is able to sensitize reactivated cells to cell death, subsequently reducing translation-competent viral reservoirs. To identify the mechanism by which cell death sensitization is occurring, we took advantage of our previously published RNAseq dataset from CD4 T cells treated with HODHBt [49]. Reactome analysis of the differentially expressed genes induced by HODHBt identified "TP53 regulates transcription of cell death

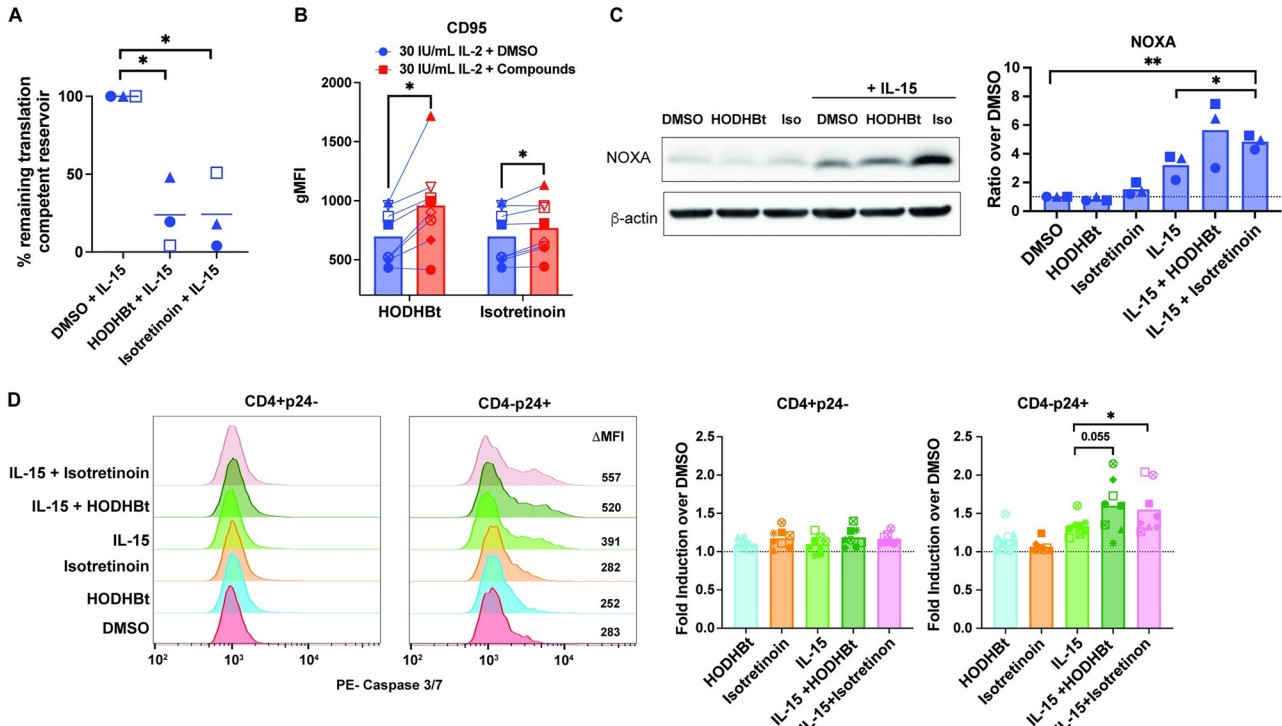

**Fig 4. Isotretinoin increases levels of pro-apoptotic proteins FAS and Noxa and increases caspase-3 activation in HIV-infected cells.** A: HIV-1 reactivation in latently infected $T_{CM}$ (n = 3) previously reactivated at day 17 with 100ng/mL IL-15 alone, IL-15 with 100$\mu$M HODHBt, IL-15 with 10$\mu$M Isotretinoin. Second reactivation from days 24–26 with $\alpha$CD3/CD28. Paired t-test was used to calculate p values (*p < 0.05; **p < 0.01). B: CD95 MFI of matched cultured uninfected $T_{CM}$ (n = 9) was measured by flow cytometry after treatment with 30 IU/mL IL-2 or IL-2 plus 100$\mu$M HODHBt or 10$\mu$M of indicated compounds for 72 hours. Significant p values over cells treated with IL-2 alone. C: NOXA levels were measured in cultured uninfected $T_{CM}$ treated with 100$\mu$M HODHBt or 10$\mu$M Isotretinoin alone, plus 100 ng/mL IL-15 (n = 2–3). Paired t-test was used to calculate p values (*p < 0.05; **p < 0.01). D: Caspase-3/7 activation measured in uninfected (CD4+p24-) and infected (CD4-p24+) CD4 T cells by flow cytometry after treatment for 48 hours with 100$\mu$M HODHBt or 10$\mu$M Isotretinoin alone, plus 100 ng/mL IL-15. Histograms represents uninfected (CD4+p24-) and infected (CD4-p24+) populations in one representative donor. Graphs indicate fold induction of PE-Caspase 3/7 MFI values for treatments over DMSO controls for both uninfected (CD4+p24-) and infected (CD4-p24+) populations (n = 8). Wilcoxon matched-pairs signed rank test was used to calculate p values (*p < 0.05; **p < 0.01).

genes" and "programmed cell death" as two significant pathways modulated by HODHBt (S9A Fig). Among the genes associated with this pathway, we identified FAS (CD95) and PMAIP1 (Noxa) as upregulated by HODHBt (S9B Fig), suggesting that HODHBt and Isotretinoin may be sensitizing infected cells to both the extrinsic and intrinsic pathways of apoptosis [60]. We first confirmed that both HODHBt and Isotretinoin enhanced the expression of CD95 (Fig 4B) and also the expression of Noxa (Fig 4C) in uninfected primary CD4 T cells. We next evaluated changes in other pro- and anti-apoptotic proteins. We observed a significant increase in the levels of the pro-apoptotic protein Bim-EL with Isotretinoin in the presence of IL-15 but not HODHBt over IL-15 alone (S9C Fig). HODHBt or Isotretinoin did not change the levels of the anti-apoptotic proteins Mcl-1, Bcl-2, or Bcl-XL in the presence or absence of IL-15 (S9D Fig).

To further evaluate whether these changes in pro-apoptotic protein expression are associated with an increase in cell death, we measured caspase-3 activation levels in HIV-negative (CD4+p24-) and HIV-positive (CD4-p24+) cells in the same culture after treatment with IL-15 in the presence or absence of HODHBt and Isotretinoin. We observed an increase in activated caspase-3 in HIV-infected cells treated with IL-15 plus HODHBt and a significant

increase after treatment with Isotretinoin plus IL-15 over IL-15 alone. Importantly, this increase in caspase-3 activation was not observed in the HIV-negative cells in the same culture, further demonstrating that HODHBt and Isotretinoin only sensitize HIV-infected cells expressing p24-Gag to cell death and not uninfected cells (Fig 4D). This data suggests that Isotretinoin, and to a lesser extent HODHBt, is counteracting the pro-survival effects of IL-15 to preferentially drive HIV-infected cells into a pro-apoptotic state resulting in caspase-3 activation and cell death.

## Discussion

In this work, we have identified that the FDA-approved drug Isotretinoin enhances IL-15-mediated HIV latency reversal and preferentially sensitizes infected cells to apoptosis through increasing caspase-3 activation in a primary cell model of HIV latency. We also showed the ability of IL-15 to reactivate the translation-competent viral reservoir as measured by both production of de-novo p24 protein expression in cell lysates and p24 release in the supernatants in cells isolated from PWH.

IL-15 and the superagonist N-803 have come to the forefront as promising LRAs for clinical use due to their ability to reactivate virus *ex vivo* from PWH [28, 51], and reactivate SIV in NHP models [61, 62]. A phase 1 clinical trial designed to test the safety of N-803 and the therapeutic impact on viral reactivation in PWH was also recently completed [36]. However, these studies measured viral reactivation as either an increase in cell-associated RNA or viral transcripts. In here, we show that IL-15 reactivates the translation-competent viral reservoir as measured by both de novo p24 production and viral protein release in a primary cell model of latency and cells isolated from PWH. Conversely to the promise of IL-15 as an LRA, the pro-survival effects have been implicated in sustaining persistent HIV infection by enhancing susceptibility of CD4+ T cells to infection in a SAMHD1-dependent manner [63], as well as promoting proliferation of CD4+CCR5+ T cells [64]. These confounding effects show that while there is promise in the use of IL-15 as an anti-HIV therapeutic, this will most likely need to be in conjunction with an agent that is able to counteract the pro-proliferative functions to avoid expanding the viral reservoir.

13-*cis*-retinoic acid, or Isotretinoin, is a Retinoic Acid (vitamin A, RA) derivative originally synthesized with the goal of treating vitamin A deficiency while mitigating associated liver toxicity [65–67]. Subsequently it was determined that Isotretinoin is an effective treatment for severe cystic acne [68] due to its sebocyte-specific ability to induce cell cycle arrest and apoptosis in a retinoic acid receptor (RAR)-independent manner [69], partially through induction of tumor necrosis factor-related apoptosis inducing ligand (TRAIL) [70, 71], and also inhibition of terminal differentiation [72]. For these reasons, Isotretinoin has been explored as a chemopreventative agent for treatment of different cancers [73], and was the intervention chosen in a recently completed clinical trial investigating the effectiveness to treat cervical tumors in women living with HIV (NCT00001073). In this work, we have shown that Isotretinoin is able to enhance the LRA activity of IL-15 in a primary cell model, as well as in *ex vivo* PBMCs from PWH who received Isotretinoin for 16 weeks. Interestingly, in cells isolated from PWH that were reactivated in the presence of both IL-15 and Isotretinoin, we observed a reduction in translation-competent viral reactivation. Similar *in vitro* results were previously reported after treatment with the RA derivative Acitretin [74], however these results were not confirmed by subsequent studies [75]. These results support further clinical investigation of Isotretinoin in combination with either IL-15 or N-803 for an HIV cure.

We demonstrated increased caspase-3 activation specifically in HIV-infected p24-Gag-expressing cells after treatment with Isotretinoin and IL-15 compared to IL-15 alone, suggesting that Isotretinoin specifically counteracts the pro-survival effects of IL-15 in HIV-infected cells and sensitize them to apoptosis. Mechanistically, Isotretinoin increases levels of the pro-apoptotic proteins CD95 and Noxa without changing levels of anti-apoptotic proteins Bcl-2, Mcl-1, leading to depletion of the inducible reservoir in a primary cell model. Further investigations are needed to fully elucidate the pathway through which Isotretinoin changes the balance of pro- and anti-apoptotic proteins in favor of a pro-apoptotic state and triggers death in HIV-infected cells. This phenomenon has already been described in studies identifying the Bcl-2 antagonist ABT-199 as able to induce cell death in infected cells by inhibiting Bcl-2 inactivation of procaspase-8 [37, 46, 47]. In addition, a class of molecules called targeted activators of cell killing (TACK) have been recently shown to specifically induce cell death in infected cells via viral protease activation [76], as well as inhibition of the CARD8 inflammasome inactivator DPP9 using Val-boroPro (VbP) [77].

There are several caveats to this study. Currently the complete mechanism of action of Isotretinoin has yet to be elucidated. We have previously shown that HODHBt enhances gc-cytokine activation of STAT5. However, a similar increased is not observed with Isotretinoin (S10 Fig). There are 9 different retinoid receptors grouped into three families each with an $\alpha$, $\beta$, and $\gamma$ subunits: retinoic acid receptor (RAR), retinoid-related orphan receptor (ROR), and retinoid X receptor (RXR) [78, 79]. These receptors are expressed to varying degrees in different cell subsets and there has been little work done to identify specifically which retinoids bind which receptors, and what functions are modulated by individual receptors or combinations in lymphocytes [80, 81]. Therefore, future studies are needed to identify which retinoid receptors are responsible for the increased IL-15-mediated latency reversal and pro-apoptotic state induced by Isotretinoin. Additionally, this can lead to the potential identification of other RA derivatives with more potent LRA and apoptosis-inducing activities in HIV infected cells, including synthetic second and third generation retinoids, such as Acitretin and Bexarotene, respectively, which have been developed for treatment of skin cancers such as cutaneous T cell lymphoma [82, 83]. ACTG A5325, from which we obtained PBMCs, was not primarily designed to measure the effect of Isotretinoin treatment on the size of the latent viral reservoir. Despite this, we observed that Isotretinoin enhances susceptibility to reactivation with IL-15, suggesting that Isotretinoin may influence the inducibility of the latent reservoir. We did not observe major changes in phenotype or the ability of IL-15 to induce immune activation, suggesting than other mechanisms may be associated with this phenomenon and further studies are warranted. We demonstrate that Isotretinoin can enhance the activation of caspase-3 only in productively infected cells as well as reduction of the latent reservoir in a primary cell model of HIV latency. We were not able to confirm whether specific caspase-3 activation was observed in reactivated cells from PWH due to the limited amount of latently infected cells found in ART-suppressed PWH. However, the reduction in p24 expression observed combined with our studies using a primary cell model of latency, support the hypothesis that after reactivation from latency, both HODHBt and Isotretinoin promote cell death of reactivated cells leading to a reduction in p24 levels observed both in the cell lysates and supernatants. Future studies using other measurements such as the intact proviral DNA assay (IPDA) [84, 85], or the quantitative viral outgrowth assay (QVOA) [86–88] will be required. However, in this study we were limited with the number of cells obtained from each participant and we prioritize to evaluate the inducibility of the latent reservoir upon Isotretinoin treatment *in vivo*. Furthermore, additional studies will be required to completely identify the pathway(s) through which Isotretinoin leads to caspase-3 activation in infected cells.

## Conclusion

In this work, we have identified an FDA-approved compound, Isotretinoin, that enhances IL-15-mediated latency reversal and specifically sensitizes infected cells to apoptosis, counteracting the pro-survival effects of IL-15. We also have shown that IL-15 is able to reactivate the translation-competent viral reservoir and stimulate production of de novo viral proteins and viral release in cells isolated from ART-suppressed PWH. Isotretinoin adds another tool to the arsenal of small molecule compounds that induce cell death of HIV-infected cells with the ultimate goal of fully eradicating the latent viral reservoir and providing a cure strategy that can be implemented globally.

## Supporting information

**S1 Fig. Correlation of CD69 surface expression with reactivation by HODHBt and derivatives.** Levels of CD69 positive cells 24 hr after treatment of cultured TCM in 5 different donors were correlated with levels of HIV-1 reactivation at 48 hr. Error bars indicate SD. Partial data from Bosque et al, Cell Reports, 2017.
(TIF)

**S2 Fig. Isotretinoin dose response.** A: CD69 normalized MFI of matched cultured uninfected TCM (n = 4) were measured by flow cytometry after treatment with a dose response of Isotretinoin in the presence of 30 IU/mL IL-2 for 72 hours. B: Normalized viability of Isotretinoin dose response from (A). Error bars indicate SEM.
(TIF)

**S3 Fig. Gating strategy for A5325 PBMC phenotyping analysis for samples from clinical trial ACTG A5325.**
(TIFF)

**S4 Fig. Gating strategy for A5325 PBMC activation analysis for samples from clinical trial ACTG A5325.**
(TIFF)

**S5 Fig. Phenotyping and Activation Results for samples from clinical trial ACTG A5325.** CD4/CD8 ratio (A), CD45RA+ subset phenotyping (B) and CD45RA- subset phenotyping (C) of PBMCs for each participant/ time point prior to reactivation stimulation. D: CD4 counts normalized to $10^6$ PBMCs plated for each participant/ time point. Wilcoxon matched-pairs signed rank test was used to calculate p values (*p < 0.05; **p < 0.01).
(TIF)

**S6 Fig. Individual reservoir size.**
(TIFF)

**S7 Fig. Individual p24 supernatant results.** Individual p24 results in supernatants for 12 ART- suppressed PWH (HIV/MST) and 2 HIV negative participants (HD). Open circles denote wells with non-quantifiable p24 measurements as described in the methods. LODs listed for respective analysis runs, output by Quanterix software. Average HIV-negative baseline (aHNB) calculated as described in methods. N.D not determined.
(TIF)

**S8 Fig. Individual p24 cell lysate results.** Individual p24 results in cell extracts for 12 ART-suppressed PWH (HIV/MST) and 2 HIV negative participants (HD). Open circles denote wells with non-quantifiable p24 values but where protein was measure via BCA as described in methods. Average HIV-negative baseline (aHNB) calculated as described in methods. N.D not

determined.
(TIF)

**S9 Fig. HODHBt upregulates genes involved in cell death in CD4 T cells.** A: Reactome analysis of pathways modulated after HODHBt treatment. B: HODHBt treatment upregulates Noxa and CD95 in CD4 T cells. C and D: Levels of anti- and pro-apoptotic proteins were measured in cultured uninfected TCM treated with $100\mu$M HODHBt or $10\mu$M Isotretinoin alone, plus 100 ng/mL IL-15 for 24 hours (n = 2–3). Paired t-test was used to calculate p values (*p < 0.05).
(TIF)

**S10 Fig. Isotretinoin does not enhance pSTAT5 in CD4 T cells.** Levels of phosphorylated STAT5 and total STAT5 were measured in cultured uninfected TCM after treatment with $100\mu$M HODHBt, $10\mu$M Isotretinoin alone or plus 100ng/mL IL-15 for 24 hours (n = 3).
(TIF)

**S1 Data. CMAP analysis.**
(XLSX)

## Author Contributions

**Conceptualization:** J. Natalie Howard, Alberto Bosque.

**Funding acquisition:** Alberto Bosque.

**Investigation:** J. Natalie Howard, Callie Levinger, Selase Deletsu.

**Methodology:** J. Natalie Howard, Callie Levinger, Selase Deletsu, Rémi Fromentin, Nicolas Chomont, Alberto Bosque.

**Project administration:** Alberto Bosque.

**Supervision:** Rémi Fromentin, Nicolas Chomont, Alberto Bosque.

**Visualization:** J. Natalie Howard, Alberto Bosque.

**Writing – original draft:** J. Natalie Howard, Alberto Bosque.

**Writing – review & editing:** J. Natalie Howard, Rémi Fromentin, Nicolas Chomont, Alberto Bosque.

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
