## [Decision Letter · Decision Letter 0]

30 Jul 2024

Dear PhD Bosque,

Thank you very much for submitting your manuscript "Isotretinoin promotes elimination of translation-competent HIV latent reservoirs in CD4T cells" for consideration at PLOS Pathogens. As with all papers reviewed by the journal, your manuscript was reviewed by members of the editorial board and by several independent reviewers. The reviewers appreciated the attention to an important topic. Based on the reviews, we are likely to accept this manuscript for publication, providing that you modify the manuscript according to the review recommendations.

Sincerely,

Daniel C. Douek

Academic Editor

PLOS Pathogens

Richard Koup

Section Editor

PLOS Pathogens

Michael Malim

Editor-in-Chief

PLOS Pathogens

orcid.org/0000-0002-7699-2064

Reviewer Comments (if any, and for reference):

Reviewer's Responses to Questions

**Part I - Summary**

Reviewer #1: The persistent challenge in curing HIV lies in the presence of latent viral reservoirs within CD4+ T cells and other cell types. Antiretroviral therapies (ART) can suppress HIV but cannot eliminate these reservoirs, which remain undetected by the immune system. The "shock-and-kill" strategy, which reactivates latent HIV to induce cell death, is a promising approach. Despite the potential of latency-reversing agents (LRAs) like IL-15 and its superagonist N-803 to reactivate latent HIV, their effectiveness is limited by the survival of reactivated cells. This study aims to identify FDA-approved compounds that can enhance LRA activity and sensitize HIV-infected cells to apoptosis.

Using the Connectivity Map (CMap), the researchers identified isotretinoin, a retinoic acid derivative, as a compound that induces similar transcriptional changes as HODHBt, a previously identified small molecule by the Bosque group. Isotretinoin enhances IL-15-mediated latency reversal without promoting proliferation of memory CD4 T cells. In ex vivo analyses of PBMCs from ART-suppressed individuals treated with isotretinoin, the combination of IL-15 and isotretinoin significantly increased latency reversal. The combination also increased caspase-3 activation, leading to apoptosis specifically in HIV-infected cells, suggesting isotretinoin's potential to reduce translation-competent HIV reservoirs. This is an exciting observation.

The study shows that isotretinoin increases the expression of pro-apoptotic proteins like CD95 (FAS) and Noxa in HIV-infected cells while maintaining low levels of anti-apoptotic proteins. This shift in the balance towards apoptosis leads to the activation of caspase-3, resulting in the death of reactivated HIV-infected cells. This mechanism is distinct from that of HODHBt, which enhances IL-15-mediated signaling through sustained STAT5 phosphorylation.

The clinical trial ACTG A5325 assessed isotretinoin's effects on CD4 counts and the latent viral reservoir in ART-suppressed individuals. Phenotypic characterization of PBMCs showed no significant changes in CD4/CD8 ratios or T cell subsets between pre- and post-treatment in the isotretinoin group. However, cells from isotretinoin-treated individuals exhibited increased susceptibility to IL-15-mediated latency reversal ex vivo. The data suggest that isotretinoin enhances the inducibility of the latent reservoir without exacerbating immune activation or proliferation of memory CD4 T cells.

The findings indicate that isotretinoin, in combination with IL-15, represents a potentially exciting, novel therapeutic approach to target and eliminate translation-competent HIV reservoirs. The study supports further clinical investigation of isotretinoin in combination with IL-15 or N-803 to enhance latency reversal and promote the apoptosis of reactivated HIV-infected cells. This approach adds to the arsenal of strategies aimed at eradicating latent HIV and achieving a functional cure.

Isotretinoin enhances IL-15-mediated HIV latency reversal and sensitizes infected cells to apoptosis, providing a promising strategy for reducing the latent HIV reservoir. This study identifies isotretinoin as a potential component of future therapeutic interventions aimed at curing HIV by eliminating latent reservoirs and promoting the death of reactivated HIV-infected cells. Although further research is needed to fully elucidate the pathways involved and to optimize the combination of isotretinoin with other therapeutic agents, the observations presented here have the clear potential to impact future therapies. The conclusions in the manuscript by Howard and co-workers stem from a wealth of novel data and observations that will no doubt help advance the field.

Reviewer #2: In this very interesting manuscript, J. Natalie Howard and colleagues show that isotretinoin synergizes with IL-15 to augment expression of HIV p24, induces preferential activation of caspase 3/7 in p24-expressing cells, and reduces the inducible translation-competent reservoir in a primary cell model of latency. In addition, they show that in vivo treatment with isotretinoin (in a clinical trial) may augment the ability of IL-15 to augment expression of HIV p24 ex vivo. The manuscript has multiple strengths, including: 1) use of multiple experimental systems, including a primary cell Tcm model of latency, ex vivo treatment of cells from PWH, and ex vivo treatment of cells from a clinical trial; 2) measurement of HIV p24 expression using an ultrasensitive assay, as well as measures of viability, “activation” (CD69), and expression of different human proteins involved in cell death; 3) inclusion of appropriate controls; and 4) use of an FDA-approved drug with potential clinical relevance. Their findings add new data to several prior publications that disagreed about whether retinoids can induce preferential killing of HIV-infected cells, and their data suggest some new mechanistic insights. At the same time, some study limitations should be discussed, such as why the EC50 was measured using CD69 expression (instead of HIV expression or killing), whether the isotretinoin dose of 10uM can be achieved in vivo, and why the results from Fig 2 and Fig 3 seem to contradict each other. In addition, there are multiple areas of the text that would benefit from some textual revision. It would also be interesting to see how isotretinoin and IL-15 affect HIV DNA and RNA levels, although these are not essential for the manuscript, which already includes a lot of data.

**Part II – Major Issues: Key Experiments Required for Acceptance**

Reviewer #1: (No Response)

Reviewer #2: (No Response)

**Part III – Minor Issues: Editorial and Data Presentation Modifications**

Reviewer #1: (No Response)

Reviewer #2: Line 40-42: There have been cases of sterilizing cure after SCT from a donor who was not CCR5d32, and also from an individual who received no SCT.

83-85: Here, they could cite Kadiyala GN et al, AIDS, 2024.

104-105: This sentence is somewhat confusing. You make it sound as if HODHBt stimulates the gc-cytokine pathway, but later on, you say that it does little unless it is combined with a gc-cytokine.

113-115: Is it fair to compare gene expression changes in primary cells and cell lines?

120-126: While the authors have previously shown an association between CD69 expression and levels of both pSTAT5 and reactivation, there might not be any causal connection. Thus, the relevance of CD69 here is somewhat unclear. Moreover, CD69 has been suggested to be both an early marker of T cell activation and a marker of tissue retention.

Fig S2: What is the “normalized viability?”

126-128: The EC50 was calculated based on induction of CD69, which is not the most relevant marker. Did you measure the EC50 for any changes in HIV expression or killing of infected cells?

128-130: The concentration of isotretinoin – 10uM – seems high. Can this concentration be achieved in vivo?

142-143 and Fig 1E: When you say “uninfected,” do you mean cells not exposed to virus? By latently infected, do you mean the virus-exposed cells, which would consist of a mixture of uninfected and latently infected?

Fig 1E: To assess the effect of isotretinoin on viability, it looks like the isotretinoin+IL-15 was compared between the uninfected and latent cells. It seems like it would be better to compare the same cell type that was or was not exposed to isotretinoin (i.e.: DMSO+IL-15 vs. isotretinoin+IL-15).

162-165: Why did placebo increase Tcm and decrease Ttm and Tem?

171-173, Fig 2A: Did you look at other markers of activation, such as HLA-DR+CD38? Rather than compare the pre- and post-placebo, it seems like it would be better to compare not just the pre- and post-isotretinoin (as you did), but also to compare the post-placebo and post-isotretinoin.

174-177: The way this sentence is written, it is not clear that both of these observations came from the placebo arm. Perhaps you should say, “In the placebo arm, we saw a significant increase in both…” Why do you see these changes after placebo?

177-180: If not already done, it would be best to compare post-isotretinoin to post-placebo.

Fig 2B: Here, it would be helpful to show the median values. In the cells from pre- and post-placebo, was there any difference between the unstimulated and IL-15 conditions? In the post-placebo, was there any difference between the unstimulated and anti-CD3/CD28? In the pre-isotretinoin, was there a difference between the unstimulated and IL-15? I would expect differences.

Fig 2C: In addition to comparing the pre- and post-, you should also compare the post-placebo to post-isotretinoin.

198-199: Did you measure viability directly, in addition to measuring LDH in supernatant?

207-209 and Fig 3: The results from Fig 3 seem to contradict those from Fig 2. Both experiments were performed ex vivo, and both were analyzed at the same time point. In Fig 2, the data suggest that cells from isotretinoin-treated PWH show an increase in p24 after ex vivo treatment with IL-15. However, in Fig 3A-B, ex vivo treatment with isotretinoin+IL-15 resulted in significantly less p24 in the supernatant than IL-15 alone, and there was a similar trend in the cell lysates. I understand the authors’ explanation of Fig 3 - that isotretinoin+IL-15 is leading to killing of p24+ cells - but it could also be that ex vivo treatment with isotretinoin actually reduces the effect of IL-15 on increasing p24 expression, or that both are true. Conversely, if ex vivo treatment with isotretinoin increases the killing of p24+ cells after IL-15 exposure (Fig 3), why does in vivo treatment with isotretinoin not lead to a similar decrease in p24 after ex vivo treatment with IL-15 in Fig 2? Could it be that the effect of isotretinoin to augment death of p24+ cells is not observed in vivo? To test your theory, can you do a time course experiment where you take cells from the isotretinoin trial, treat ex vivo with IL-15, and show that the p24 first goes up and then comes down over time? Alternatively, can you measure HIV DNA over time or compare to negative control to show killing of infected cells?

220-222: As above, another explanation is that isotretinoin actually reduced the ability of IL-15 to induce p24 expression ex vivo.

231-233 and Fig 4A: To clarify, is the data shown from day 24-26, after activation with anti-CD3/CD28? Why use the paired T test instead of the Wilcoxon?

242-244: Based on Fig 4, isotretinoin did not seem to increase expression of Noxa unless it was combined with IL-15.

250-253 and Fig 4D: Rather than saying “HIV-negative” and “HIV-positive,” I would say p24- and p24+. The term HIV-negative suggests the cells are not infected, but some may be infected and not expressing p24.

255-258: By “HIV-negative,” I’m guessing you mean p24-?

257-258: You should say that isotretinoin only sensitizes HIV Gag-expressing cells to cell death.

300: Instead of HIV-infected, I would say HIV p24-expressing.

328-330: Did prior publications from the isotretinoin trial show any data about levels of total HIV DNA, intact/defective HIV DNA, or IUPM from QVOA? If so, it would be helpful to remind the readers of these results in the Intro.

440: Do the activating beads fall off before 96h? If not, why did you do the magnetic separation? If the beads did not fall off, I would imagine that some cells (including many of the activated ones) would remain stuck to the magnet, while other cells (including many that did not get activated) would be removed in the supernatant. Which cells were lysed and analyzed? Was it the non-magnetic ones? Why not measure p24 in all of the cells together?

441: Which cell pellet – the magnet bound cells, or the non-magnetic ones?

452-455: This sentence is confusing.

PLOS authors have the option to publish the peer review history of their article (what does this mean?). If published, this will include your full peer review and any attached files.

Reviewer #1: No

Reviewer #2: No

Figure Files:

Data Requirements:

Reproducibility:

References:

---

## [Decision Letter · Decision Letter 1]

18 Sep 2024

Dear PhD Bosque,

We are pleased to inform you that your manuscript 'Isotretinoin promotes elimination of translation-competent HIV latent reservoirs in CD4T cells' has been provisionally accepted for publication in PLOS Pathogens.

Best regards,

Daniel C. Douek

Academic Editor

PLOS Pathogens

Richard Koup

Section Editor

PLOS Pathogens

Michael Malim

Editor-in-Chief

PLOS Pathogens

orcid.org/0000-0002-7699-2064

Reviewer Comments (if any, and for reference):

Reviewer's Responses to Questions

**Part I - Summary**

Reviewer #2: The authors have provided satisfactory answers to my questions and have made appropriate revisions.

**Part II – Major Issues: Key Experiments Required for Acceptance**

Reviewer #2: None

**Part III – Minor Issues: Editorial and Data Presentation Modifications**

Reviewer #2: None

PLOS authors have the option to publish the peer review history of their article (what does this mean?). If published, this will include your full peer review and any attached files.

Reviewer #2: No

---

## [Editor Report · Acceptance letter]

26 Sep 2024

Dear PhD Bosque,

We are delighted to inform you that your manuscript, "Isotretinoin promotes elimination of translation-competent HIV latent reservoirs in CD4T cells," has been formally accepted for publication in PLOS Pathogens.

Best regards,

Michael Malim

Editor-in-Chief

PLOS Pathogens

orcid.org/0000-0002-7699-2064